# Quantifying Coexistence Concentrations in Multi-Component Phase-Separating Systems Using Analytical HPLC

**DOI:** 10.3390/biom12101480

**Published:** 2022-10-14

**Authors:** Anne Bremer, Ammon E. Posey, Madeleine B. Borgia, Wade M. Borcherds, Mina Farag, Rohit V. Pappu, Tanja Mittag

**Affiliations:** 1Department of Structural Biology, St. Jude Children’s Research Hospital, 262 Danny Thomas Place, Memphis, TN 38105, USA; 2Department of Biomedical Engineering, Center for Biomolecular Condensates (CBC), James McKelvey School of Engineering, Washington University in St. Louis, St. Louis, MO 63130, USA

**Keywords:** phase separation, biomolecular condensates, coexistence line

## Abstract

Over the last decade, evidence has accumulated to suggest that numerous instances of cellular compartmentalization can be explained by the phenomenon of phase separation. This is a process by which a macromolecular solution separates spontaneously into dense and dilute coexisting phases. Semi-quantitative, in vitro approaches for measuring phase boundaries have proven very useful in determining some key features of biomolecular condensates, but these methods often lack the precision necessary for generating quantitative models. Therefore, there is a clear need for techniques that allow quantitation of coexisting dilute and dense phase concentrations of phase-separating biomolecules, especially in systems with more than one type of macromolecule. Here, we report the design and deployment of analytical High-Performance Liquid Chromatography (HPLC) for in vitro separation and quantification of distinct biomolecules that allows us to measure dilute and dense phase concentrations needed to reconstruct coexistence curves in multicomponent mixtures. This approach is label-free, detects lower amounts of material than is accessible with classic UV-spectrophotometers, is applicable to a broad range of macromolecules of interest, is a semi-high-throughput technique, and if needed, the macromolecules can be recovered for further use. The approach promises to provide quantitative insights into the balance of homotypic and heterotypic interactions in multicomponent phase-separating systems.

## 1. Introduction

Phase separation is a biophysical process in which a macromolecular solution separates spontaneously into two coexisting phases. The two phases are a dense phase enriched in macromolecules with a macromolecular concentration *c*_dense_, and a dilute phase deficient in macromolecules with a macromolecular concentration *c*_sat_. In a binary mixture comprising macromolecules of a specific type dissolved in a complex solvent, the separation into two coexisting phases occurs at and above a system-specific threshold macromolecular concentration denoted as *c*_sat_. At concentrations below *c*_sat_, the solution is stable as a one-phase system. Over the last decade, evidence has accumulated that phase separation, which is a segregative transition, can be used to explain the observed compartmentalization of cellular matter [1,2,3,4]. Phase separation is thought to contribute, at least in part [5], to the formation of membraneless compartments known as biomolecular condensates [2]. These include condensates such as nucleoli [6,7] in the nucleus, and stress granules [8,9,10] and P bodies [11] in the cytoplasm. Phase separation has also been implicated in the formation of DNA repair foci [12,13], transcription centers [14,15,16] and membrane receptor clusters [17,18].

The ubiquitous roles invoked for phase separation in cells suggest that its dysregulation can result in disease states [19,20]. Indeed, evidence is accumulating that cancer pathogenesis can be mediated by either the abrogation of functional condensates through mutations of scaffolding molecules [21], or by the creation of aberrant condensates through the fusion of phase separating molecules with effector domains via chromosomal translocations [22,23,24]. Aberrant maturation of condensates is also thought to underlie the pathogenesis of a spectrum of neurodegenerative diseases [25,26,27,28,29,30].

Understanding the mechanisms of phase separation and quantitative comparisons of the system-specific driving forces for phase separation requires the quantitative characterization of phase-separating systems. Of particular interest is the contribution of multiple macromolecular components to phase separation [31], and how ligands influence the phase behavior [32,33]. Semi-quantitative methods for mapping phase diagrams include titrating a series of input concentrations of the constituent components and determining the presence or absence of phase separation via microscopy or turbidity measurements. This creates a grid, that yields an approximation of the low concentration arm of a phase boundary, where the number of input concentrations that are titrated and the concentration ranges they help explore will determine the accuracy and resolution of the inferred phase boundary [21,26,34]. These semi-quantitative phase boundaries have proven very useful in determining some key features of biomolecular condensates [9]. However, these methods often lack the precision necessary to be used for generating quantitative models [35,36]. Such physics-based models can provide insight into the underlying interactions by extracting thermodynamic parameters such as the critical temperature, free energy of mixing, and interaction strengths [35,36,37,38,39]. In addition, accurately determining the coexisting dilute and dense phase concentrations in multi-component systems provides information on the underlying contributions from homo- vs. heterotypic interactions [31,40]. For example, while two ligands may have similar observable effects on a condensate in terms of their partition coefficients, the underlying molecular mechanisms and effects on the driving force for phase separation may be very different [32,33]. Only with quantitative measurements over a range of input concentrations is it possible to distinguish underlying mechanisms of the modulation or regulation of phase behavior. Hence, there is a clear need for techniques that allow quantitation of coexisting dilute and dense phase concentrations of phase-separating biomolecules, especially in systems with more than one type of macromolecule. While quantitative methods have been developed for single-component systems [41,42], accurate determination of the coexistence concentrations of all species in multi-component systems is still a challenge.

Here, we report the development and deployment of analytical High-Performance Liquid Chromatography (HPLC) to separate and quantify distinct biomolecules and thereby access dilute and dense phase concentrations needed to reconstruct coexistence curves in multicomponent mixtures in vitro. This approach proves to be suitable because: (i) it can separate several components in multi-component systems, (ii) it is a label-free technique, (iii) through the choice of a suitable input volume it can frequently detect lower amounts of material than can be accessed using classic UV-spectrophotometers, (iv) it is applicable to a broad range of macromolecules of interest (e.g., nucleic acids, proteins, sugars), (v) it is a semi-high-throughput technique in that ca. 200 samples can be queued for measurement with the instrumentation used in this work, and finally (vi) if needed, the macromolecules can also be recovered for further use by coupling the HPLC instrument to a fraction collector.

To illustrate the utility of the HPLC approach, we deploy it to determine the saturation concentrations for a single-component phase-separating system and show their agreement with previously measured coexistence concentrations determined via classic UV-spectroscopy measurements. We then demonstrate that this method enables the determination of concentrations of each of the components in the dilute and dense phases of a two-component phase-separating system. This, as will be discussed in detail in a separate contribution [43], can provide access to information regarding the slopes of tie lines. The HPLC method that we introduce here has the potential to provide quantitative insights into the interplay between homo- and heterotypic interactions in multi-component systems and help further our understanding of the driving forces for phase separation in biochemical reconstitutions of the phase behaviors of complex macromolecular mixtures.

## 2. Results

### 2.1. Rationale for the Proposed Approach

Accurate dilute and dense phase concentrations in single-component phase-separating systems can be determined by absorbance measurements of the dilute phase after centrifugation to pellet the dense phase [35,36,41]. If the saturation concentration is low, or if samples have low extinction coefficients, the absorbance values may be below the reliable detection limit of the spectrophotometer. Furthermore, in multi-component phase-separating systems, it is non-trivial to quantify the concentrations of individual components. If the sample contains more than one protein, UV absorption at 280 nm cannot be used to separate the contributions from the two proteins. Likewise, in systems containing protein and nucleic acids, the overlapping absorption spectra cannot be reliably deconvoluted. Labelling components with spectroscopically resolvable fluorophores and determining their concentrations in the dense phase from the fluorescence intensity is an option, but many controls are needed for accurate measurements [35]. In turn, fluorophores may influence the phase behavior. Large fluorophores, like GFP, may have a dramatic effect on the solubility [44,45], but even small fluorophores may be able to perturb phase behavior as they are often charged and have aromatic moieties which are key determinants of phase behaviors of many phase-separating biomolecules [10,31,35,36,37,39,44,46].

As mentioned previously, it is possible to generate semi-quantitative estimates of the locations of phase boundaries by titrating the components and then using turbidity or microscopy to determine the presence/absence of droplets. These methods, however, are semi-quantitative, only yielding estimates of saturation concentrations because concentrations are evaluated in a stepwise fashion. They also do not provide access to tie lines. For microscopy, there is the additional concern that interactions (or lack thereof) of condensates with the slide surface can interfere with observation, and surfaces may need to be functionalized and optimized separately for mutants of the biomolecules. Given these complications, using a label-free method that relies on the intrinsic properties of the native molecules would be preferable.

### 2.2. Details of the Approach 

Combining the established approach of separating dilute and dense phases via centrifugation with the use of analytical HPLC to separate and quantify sample components yields a semi-high-throughput, robust, and highly quantitative method (Figure 1). First, a column needs to be selected and tested to confirm that it can separate the components (see “General considerations for implementation of the method”). Then, separate standard curves for each component are determined by making several injections of known concentration (*c_A_*) and volume (*V_A_*) and integrating the peak from the chromatogram (*I_A_*). This generates a standard curve for each species (example given in Figure 2A), which is fit to Equation (1)
(1)$$I_{A}=slope\times n_{A}+intercept$$
to determine the slope and intercept for the given component. Here, *n_A_ = c_A_*
×
*V_A_*, the amount of biomolecule in moles. This yields the input concentration *c_A_* for a given volume *V_A_*. With the resulting standard curve, *c*_sat_ and *c*_dense_ for the component can be determined from injections of appropriate samples as follows, and schematically shown in Figure 1. After preparation and incubation of the phase-separating sample, the dilute and dense phases are separated by centrifugation. A sample of the dilute phase is removed, injected onto the HPLC and eluted with the same gradient as used for the calibration measurements. The amount of each component present in the sample is quantified by integration of the relevant peak (*I_A_*) such that the initial sample concentration of each component (*c_A_*) is computed based on their individual standard curves and the volume of sample injected (*V_A_*). Dense phase concentrations are determined in the same way; this requires the preparation of a dense phase sample that is large enough to remove a defined volume, which is then diluted appropriately for HPLC processing.

### 2.3. Validation of the Method with a Single-Component Phase-Separating System

To test the capabilities of analytical HPLC for determining coexisting dilute and dense phase concentrations, we determined the coexisting concentrations of phase-separated samples of the low-complexity domain (LCD) of hnRNPA1, hereafter referred to as A1-LCD. We previously determined the concentrations of coexisting phases by UV absorption [35,41]. The standard curve shows a linear relationship between input sample amount and peak area allowing the use of linear regression analysis to determine sample concentrations (Figure 2A). In all cases, *c*_sat_ and *c*_dense_ that we estimated from the HPLC chromatograms agree with the previously reported values (Figure 2B) [35]. Thus, the method appears to be suitable for the quantitative determination of coexisting dilute and dense phase concentrations of biomolecules.

### 2.4. Application to Multi-Component Phase Separation

Having established the functionality of analytical HPLC in a single-component phase-separating system, we next employed the approach to quantify the saturation concentration of a Gcn4 construct (for details see Methods) in the presence of polyethylene glycol with an average molecular weight of 8 kDa, referred to hereafter as PEG8000. The absorbance spectrum for PEG8000 partially overlaps that of Gcn4 at 280 nm, and thus its detection must be performed separately from Gcn4 in order to determine the dilute phase concentration of Gcn4. Before using HPLC to quantify the concentrations of multiple components in a sample, it needs to be confirmed that the components elute separately and with sufficient resolution. This was shown to be case, thus allowing for accurate assessments of the concentration of the Gcn4 component. We quantified the *c*_sat_ values of Gcn4 as a function of different input concentrations of PEG8000. As expected, the concentration of Gcn4 in the coexistent dilute phase decreases with increasing concentrations of PEG8000 (Figure 3A). This implies that for the Gcn4 system, PEG8000 behaves mostly like a crowder that enhances the driving forces for phase separation through depletion mediated attractions, which refers to the enhanced inter-Gcn4 attractions that most likely arises from exclusion of the crowder from the dense phase. 

Next, we investigated a two-protein system consisting of A1-LCD and the FUS prion-like domain (FUS-PLD); the two proteins can phase separate on their own. However, they also form a single dense phase when mixed. The mixture of the A1-LCD and FUS-PLD is a ternary system comprising two macromolecules in a solvent. To study phase separation in this mixture, we need to be able to measure the concentrations of both macromolecules in the coexisting dense and dilute phases in the scenario where the ternary system separates into two coexisting phases. We quantified the dilute and dense phase concentrations for both protein components at a single mixing ratio and concentration. The chromatogram in Figure 3B shows that A1-LCD and FUS-PLD elute separately, allowing for integration of the peaks and determination of dilute and dense phase concentrations of each component; the results are shown in Figure 3C. The saturation concentrations of separate A1-LCD and FUS-PLD solutions are higher than the coexisting dilute phase concentrations of each of the components in the mixture. In fact, even the sum of the dilute phase concentrations in the mixture is lower than the *c*_sat_ values measured for either system in a binary mixture comprising just one type of macromolecule and the solvent. We also determined dense phase concentrations for the A1-LCD/FUS-PLD mixture and the tie line (Figure 3C,D), which provides insights into contributions from homo- and heterotypic interactions to phase separation [40]. A detailed discussion of the complex phase behaviors of A1-LCD and FUS-PLD mixtures will be presented elsewhere [43]. 

We have demonstrated that determination of dilute and dense phase concentrations of coexisting phases can be achieved by integrating peaks in HPLC chromatograms. The reliability of the measurements is established via favorable comparisons to estimates obtained using measurements based on established techniques [41]. We can go beyond the study of binary mixtures comprising just one type of macromolecule and leverage the ability to separate a system of multiple components using HPLC to determine concentrations of more than one type of macromolecule in coexisting phases. This information gives access to coexisting dilute and dense phase concentrations in a multi-component phase boundary, from which tie lines can be determined. This provides powerful information that is difficult to procure in other ways and can be used to dissect contributions from homo- and heterotypic interactions to phase separation.

### 2.5. General Considerations for Implementation of the Method

To employ and adapt the approach described in this study, the following points need to be considered:

(a)Column: Columns to achieve separation include normal-phase, reverse-phase, ion exchange and size exclusion columns, which are readily available for HPLC systems. The work presented here used C4 or C18 (ReproSil Gold 200; Dr. Maisch) reverse-phase columns. (b)Mobile phase: The mobile phase solvents used are primarily dictated by the column. The typical chromatographic buffers used for size exclusion and ion exchange are aqueous buffers, while RP-HPLC uses a gradient of organic solvents in water. However, also within the remit of RP-HPLC, there are different options possible for the organic solvent, including acetonitrile, methanol, and tetrahydrofuran. The solvents used must be miscible with water and of HPLC-grade quality to minimize their contribution to the absorbance signals measured. In this work, gradients used involved the mixing of H_2_O + 0.1% TFA (trifluoroacetic acid) with pure acetonitrile. 0.1% TFA yields a pH of 2.1 ensuring full ionization of analytes and acts as a weak ion-pairing agent thereby conferring more uniform binding of each analyte. This yields sharper peaks and more reproducible elution profiles. Use of TFA in just water, and not in the acetonitrile, is employed as it effectively adds an ion exchange component to the RP-HPLC separation and can result in better peak separation. Of note, the low pH results in the denaturation of protein structure. In cases where it is desirable to recover the components, reverse-phase columns are only suitable if the macromolecules readily refold. (c)Gradient: Optimization of the gradient is required to obtain sufficient separation between eluting species. It is important to consider sufficient equilibration time given the column volume if step changes are made at any point in the overall gradient run. The appropriate combination of points a, b and c is key to a successful use of the HPLC methodology and likely requires iteration for optimization based on the types of samples that are being studied.(d)Detection: HPLC systems may have different detection capabilities ranging from a single absorbance wavelength to setups with photodiode array detectors providing absorbance spectra rather than single wavelengths, or even fluorescence detectors. This work made use of an HPLC system with a dual-selectable wavelength detector. The selection of wavelengths to be monitored will depend on the macromolecule of interest. Typical choices include 280 nm for proteins containing aromatic residues, 260 nm for nucleic acids and 230 to 215 nm for proteins lacking aromatic residues. The monitored wavelength should also avoid interference from solvent components.(e)Column loading: The range of volumes that can be injected onto the column will depend on the system at hand. Injection of accurate volumes, a prerequisite for accurate determination of coexistence concentrations, is most easily achieved with an autoinjector. Further, the amount of macromolecule of interest in the sample should yield an absorbance signal in the linear range of the detector as confirmed through the standard curve. The amounts for which this can be achieved will vary based on the extinction coefficient of the molecule, the wavelength being monitored, and the width of the elution peak, which can be optimized by solvent choice and gradient properties. A further consideration is that loading of high concentrations of some buffer components such as glycerol or PEG can lead to contamination and ultimately damage the column. (f)Washing: It is good practice to perform wash programs/cycles between batches of samples to ensure that the column remains in good working order and is frequently cleaned. This avoids material or contamination from previous runs interfering with following measurements. (g)Tests: Routine running of blank injections using the method gradient is valuable to check that sample material has not been retained on the column. Retention in the column can lead to subsequent elution that interferes with the quantitation of components in injected samples.(h)Sample recovery: If the HPLC system is coupled to a fraction collector, the eluted peaks can be collected to recover sample components. In the case of RP-HPLC, these fractions are best dried on a speed-vac and then resuspended in the buffer of choice. Keep in mind however, that as RP-HPLC denatures the protein, structured proteins need to be refolded.
*Considerations regarding handling of dense phase:*
(i)Viscosity of dense phase: The dense phase is highly viscous and needs to be carefully pipetted. We recommend the use of a positive displacement pipette to minimize errors and achieve accurate volumes (see also [41]. The variability in the measured dense phase concentrations is higher than the measured dilute phase concentrations as can be seen in Figure 2B and Figure 3C,D, but the percentage errors are relatively small. Compared to error sources in other approaches for determining dense phase concentrations, e.g., microscopic determination of fluorescence intensity in the dense phase, the error contribution from pipetting the viscous dense phase is relatively small and manageable. Several replicate measurements should be performed to get a sense of their precision.(j)Sample requirements: The required biomolecule amounts to generate sufficient dilute phase for detection depend almost exclusively on the extinction coefficient of the biomolecule. Dense phase requirements can be more limiting. We typically remove 2 μL of dense phase for dilution and subsequent injection into the HPLC. The amount of protein needed to generate a slightly larger volume of dense phase depends on the dilute vs. dense phase concentrations and the concentration of the stock solution. If we, e.g., consider the hnRNPA1 LCD (Figure 2B) with dilute and dense phase concentrations at 20 °C of ~100 μM and ~20 mM, a stock solution used to generate a dense phase sample could be 100 μL of a 1 mM protein. Induction of phase separation (by addition of NaCl to 150 mM final concentration in this experiment) would result in approximately 95.5 μL of 100 μM dilute phase and 4.5 μL of 20 mM dense phase. Notably, the resulting dense phase volume is not only determined by the total amount of protein but also by how far above the saturation concentration the preparation starts, with higher concentrations capturing a larger fraction of protein in the dense phase. Less concentrated dense phases require substantially lower protein amounts.

## 3. Materials and Methods

### 3.1. Details of Protein Constructs

Three different proteins, namely A1-LCD and FUS-PLD, which are prion-like disordered domains of the Fused in Sarcoma (F), Ewing Sarcoma (E), and Taf15 (T) (i.e., FET) family of proteins, and a short version of the canonical yeast transcription factor Gcn4, were expressed in *E. coli* and purified. A1-LCD was expressed as detailed in reference [8], and the same cloning, expression and purification strategy was employed for FUS^1−214^ (UniProt: P35637) and Gcn4. The variant of Gcn4 spans the central activation domain (residues 101–141) from *S. cerevisiae* (UniProt: P03069) connected by a short (GS)_4_-linker to the DNA-binding domain of Gcn4 (residues 222–281).

### 3.2. Phase Separation Assay

Phase separation of A1-LCD and FUS-PLD, respectively, was induced by adding NaCl to 150 mM in 20 mM HEPES (pH 7.0). Phase separation of Gcn4 was induced by titrating PEG8000 from 2.5% to 15% in 20 mM HEPES (pH 7.3), 150 mM potassium acetate, 2 mM DTT. For the multi-component A1-LCD/FUS-PLD system, 1.1 mg/mL A1-LCD was mixed with 1.1 mg/mL FUS-PLD in 20 mM HEPES (pH 7.0), 150 mM NaCl. The samples were incubated at the desired temperature for 20 min, then centrifuged at this temperature for 5 min at 12,000 rpm to separate the dilute and dense phases. Known amounts of dilute and dense phases were removed. The dense phase volume was diluted into a defined volume of 6 M GdmHCl as needed. Aliquots of the separated phases were then applied to the HPLC to determine the concentrations. 

### 3.3. HPLC instrumentation, Columns, and Solvents

The dilute (*c*_sat_) and dense phase (*c*_dense_) concentrations were determined on a HPLC instrument with UV/Vis Detector. Samples were run on a Waters HPLC system with an Autosampler (Waters 2707), a Binary HPLC Pump (Waters 1525) and a dual-channel UV/Visible Detector (Waters 2489). The wavelengths monitored were 280 nm and 230 nm. Monitored wavelengths should be chosen to avoid any interference from solvent components. ReproSil Gold 200 (5 µm, 250 mm × 4.6 mm; Dr. Maisch, Germany) columns were used; C18 for protein only samples, and C4 for Gcn4 + PEG8000 samples. The solvents used were H_2_O + 0.1% TFA (Sigma-Aldrich, Saint Louis, MO, USA) and acetonitrile (Alfa Aesar, Haverhill, MA, USA).

### 3.4. Calibration of Concentration Measurements by HPLC

For each protein, a standard curve was measured by injecting 5–6 different volumes (*V_A_*) of solution in buffer with known concentration (*c_A_*). The integral of the elution peak (*I_A_*) was obtained with the built-in Waters Empower HPLC-software. A plot of *I_A_* vs. *n_A_*, where *n_A_* = *c_A_*
×
*V_A_*, yields the line which was fit with Equation (1) (Figure 2A) to obtain the *slope* and *intercept*. The resulting standard curve enables determination of the concentration of samples with known injection volumes and resulting peak integrals. 

### 3.5. Determination of the Dilute (c_sat_) and Dense Phase (c_dense_) Concentration Using HPLC

Quantitation of *c_sat_* was achieved by injecting known volumes of each dilute phase sample onto the HPLC. Dense phase concentrations were assessed by dilution of 2 mL of dense phase, obtained using a positive displacement pipette, with 6 M GdmHCl before loading onto the HPLC. Dilution facilitates complete loading and the denaturant prevents precipitation that may occur upon dilution with incompatible buffers. For all resulting chromatograms the amounts of the relevant components were calculated from their respective standard curves, allowing the reconstruction of the coexistence line. For all data points presented, at least three replicates were measured and averaged. The resulting values were compared to dilute and dense phase concentrations determined by UV absorption on a spectrophotometer that we have previously reported [35].

## 4. Discussion

To address the need for quantitative methods to measure phase boundaries in multi-component systems, we present an analytical HPLC method to separate and quantify multiple components in coexisting dilute and dense phases. We tested the accuracy of the HPLC method by reproducing measured dilute and dense phase concentrations of coexisting phases for the binary mixture comprising a single type of macromolecule namely, A1-LCD. We then deployed the method to study a ternary system with two macromolecular components namely, Gcn4 and PEG8000 system. We used this system because the two macromolecular components have overlapping absorption spectra. We showed that the HPLC based approach enabled the separation of both macromolecules, thus allowing us to quantify the concentration of Gcn4 in the coexisting dilute phase as function of the concentration of PEG8000. The methodology was then used to determine all four coexisting concentrations for the two-component A1-LCD/FUS-PLD phase-separating system. The four concentrations are the individual dilute phase concentrations of A1-LCD and FUS-PLD, and their coexisting dense phase concentrations. These data provide direct access to tie lines, and with additional input concentrations we can map the full coexistence curve, and use information regarding the shapes of these curves as well as the slopes of tie lines to uncover the interplay between homo- vs. heterotypic interactions—a topic that we will analyze and discuss elsewhere [43].

The basic methodology described here should be able to determine the concentrations of as many species as one can resolve on the chosen column. For increasingly complex systems, not all species may be resolvable on a single column, and future development of the methodology will center around using parallel columns with different chemistries to resolve a larger number of species. This would allow for the accurate determination of coexisting concentrations for increasingly complex systems.

For protein-RNA mixtures, additional challenges include high apparent affinities in the nanomolar or sub-nanomolar range, and therefore, they may remain bound to one another even during the HPLC run. Further, the dilute phase will likely comprise a mixture of bound and unbound species as defined by a binding polynomial. Separating the bound and unbound species would provide a fuller species characterization and is a challenge to be addressed that will also be highly relevant for systems that form pre-percolation clusters [47]. Combining the HPLC methodology with other approaches is likely to be promising in this regard.

Overall, the HPLC methodology reported here enables label-free, quantitative measurements of coexisting concentrations in complex systems at semi-high throughput. The method has the potential to further our understanding of the contributions of homotypic and heterotypic interactions and how they are encoded in the sequence of biomacromolecules.

## Figures and Tables

**Figure 1 biomolecules-12-01480-f001:**
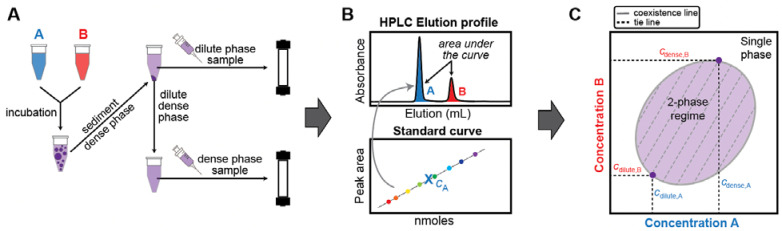
Schematic overview of the workflow used in this study to reconstruct phase boundaries via analytical HPLC. (**A**) Biomolecules A and B undergo phase separation when present at suitable concentrations and molar ratios. The sample is incubated for equilibration. Separation of the dense and dilute phases is achieved via centrifugation. Known volumes of dilute and dense phase are each separately injected onto the HPLC column and eluted with an appropriate method. (**B**) HPLC elution profile for a sample containing biomolecules A and B. Peaks are integrated and the concentrations of biomolecules A and B are determined using a standard curve. (**C**) The coexistence line and tie lines of biomolecules A and B can be reconstructed from the dilute and dense phase concentrations extracted from the elution profiles. The tie line connects the coexisting dilute and dense phase concentrations. A tie line or tie simplex is defined by its slope, and it identifies the components whose concentrations need to be constrained relative to one another to yield the concentrations of the coexisting phases.

**Figure 2 biomolecules-12-01480-f002:**
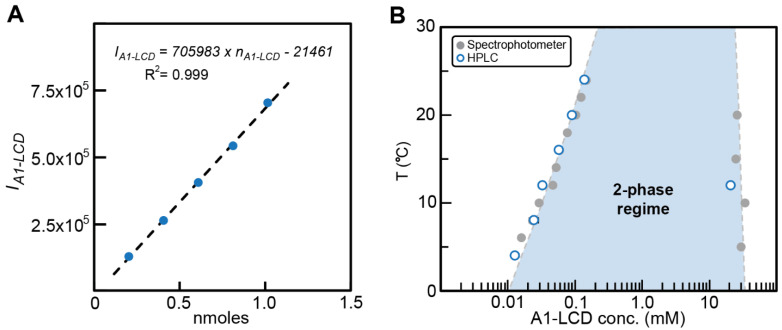
Determination of *c*_sat_ and *c*_dense_ by analytical HPLC. (**A**) The standard curve for A1-LCD was determined by plotting the area under the HPLC elution peak (*I_A1-LCD_*) from injections of different amounts of A1-LCD, *n_A1-LCD_* in nmoles. (**B**) Comparison of *c*_sat_ and *c*_dense_ values obtained via measurements using HPLC vs values from UV absorption measurements by spectrophotometer for A1-LCD as a function of temperature [36]. The dashed line represents a fit of the Flory Huggins equation to the spectrophotometer data. The shaded area thus represents the 2-phase regime determined by the coexistence concentrations and other extant data [35,36].

**Figure 3 biomolecules-12-01480-f003:**
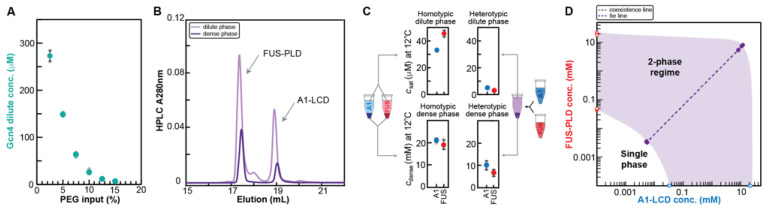
Determination of dilute and dense phase concentrations for multi-component systems. (**A**) Dilute phase concentrations of Gcn4 in the presence of increasing concentrations of PEG8000. Individual measurements are shown in grey, with the average shown in green. At least three replicates per sample were measured and error bars represent the standard deviation. (**B**) HPLC chromatogram showing the elution profile of samples of the dilute and dense phase for the A1-LCD/FUS-PLD mixture. The dense phase sample was diluted prior to injection. For a comparison of absorbance at 230 and 280 nm, please see Appendix A. (**C**) *c*_sat_ and *c*_dense_ for A1-LCD and FUS-PLD for the case of their homotypic or heterotypic phase separation. (**D**) Data points from C are shown in a 2D phase diagram. The tie line between coexisting dilute and dense phase concentrations in the heterotypic system is shown as dashed line. The 2-phase regime is approximated as shaded area as expected from few presented data points.

## Data Availability

All data are available upon request.

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
