# Peer review of "Quantifying Coexistence Concentrations in Multi-Component Phase-Separating Systems Using Analytical HPLC"

_biomolecules, 2022, doi:10.3390/biom12101480_

Round 1
Reviewer 1 Report
This is a clear, concise and useful methodology manuscript describing the use of HPLC for the quantification of the densities of components of liquid phase transitions in both the dilute and the dense phase. The need for such a tool is well justified by the authors, who note the lower precision of methods relying on dense phase assessment by turbidity or other measures as a function of component concentrations, and the concerns associated with more precise methods that require labels for concentration quantification, which may affect the phase separation process itself. The method is shown to agree with absorbance-based measurements in a simple system comprised of A1-LCD in buffer and is then used to obtain simple phase diagrams for a model system involving PEG-induced separation of Gcn4 A1-LCD and co-separation of A1-LCD and FUS-PLD. While quantitative HPLC is not a novel approach, it's application in this context is novel and the manuscript is expected to be of interest and use to others in this rapidly growing field. I have only one minor critique, below.
The dense phase was diluted into 6M GuHCl. It would be good to explain why this was done, whether it was done for all the dense phases examined, and whether this may be generally required.
Reviewer 2 Report
Review of “Quantifying coexistence concentrations in multi-component phase-separating systems using analytical HPLC,” by Bremer and Posey et al.
Summary
The paper outlines a fast and accurate approach to measuring the concentrations of macromolecular species in multi-component phase separated droplets using HPLC. The concentration of macromolecules in the dense and dilute phases is a key quantity of interest when studying phase separated droplets. The authors nicely outline the shortcomings of commonly employed techniques, particularly fluorescence microscopy. HPLC is presented as a straight forward alternative capable of accurate, quantitative measurement of macromolecular concentrations in phase separating systems, even if the phase separation involves multiple components. I appreciate the general consideration section that guides practical implementation of the method.
Recommendation
I recommend this paper for publication following a few minor comments.
Comments
o The dense phase can be difficult to pipette. Does this introduce any errors into the analysis?
o The authors report the concentration detection limits of the HPLC, but what is the concentration limit needed for centrifugation assay and transfer to the instrument? Can you comment on the potential effect of centrifugation on the instability of the dense phase?
Reviewer 3 Report
The manuscript describes the applicability of quantitative HPLC analysis of macromolecules in coexisting two-phase systems. The manuscript describes the need for HPLC methods and why they are superior to previously applied UV and other methods that did not provide separation of potentially several types of macromolecules in the coexisting phases. The authors provide a general description of the method underlying the theoretical and practical advantages. The materials and method part describes in detail the variety of HPLC columns and conditions that can be used. At the end of the manuscript, an excellent application is described in detail. One may wonder why HPLC has not previously been used for this purpose.
Author Response
We thank the Reviewer for their positive assessment of our manuscript.
Reviewer 4 Report
The manuscript entitled : Quantifying coexistence concentrations in multi-component phase-separating systems using analytical HPLC was submitted in "biomolecules". I have some questions.
In section HPLC; The author used two wavelengths (280 and 215 nm). The wavelength at 215 is near UV cutoff of TFA. Please show the chromatogram in this wavelength.
In Figure 3(b) please check X axis. I think it should be time and start at 0 min.
The topic original or relevant in the field of biomolecules.
The references are appropriate.
Figure 1a b c should be separated to 1, 2, 3 .
Comparison with other method should be added.
Conclusion should be revised.
Why this work chosen C18 reversed column.
